# Phenotypic and Safety Assessment of the Cheese Strain *Lactiplantibacillus plantarum* LL441, and Sequence Analysis of its Complete Genome and Plasmidome

**DOI:** 10.3390/ijms24010605

**Published:** 2022-12-29

**Authors:** Ana Belén Flórez, Lucía Vázquez, Javier Rodríguez, Baltasar Mayo

**Affiliations:** 1Departamento de Microbiología y Bioquímica, Instituto de Productos Lácteos de Asturias (IPLA), Consejo Superior de Investigaciones Científicas (CSIC), Paseo Río Linares s/n, 33300-Villaviciosa, Asturias, Spain; 2Instituto de Investigación Sanitaria del Principado de Asturias (ISPA), Avenida de Roma s/n, 33011-Oviedo, Asturias, Spain

**Keywords:** *Lactiplantibacillus plantarum*, lactic acid bacteria, non-starter lactic acid bacteria, adjunct cultures, dairy microbiology

## Abstract

This work describes the phenotypic typing and complete genome analysis of LL441, a dairy *Lactiplantibacillus plantarum* strain. LL441 utilized a large range of carbohydrates and showed strong activity of some carbohydrate-degrading enzymes. The strain grew slowly in milk and produced acids and ketones along with other volatile compounds. The genome of LL441 included eight circular molecules, the bacterial chromosome, and seven plasmids (pLL441-1 through pLL441-7), ranging in size from 8.7 to 53.3 kbp. Genome analysis revealed vast arrays of genes involved in carbohydrate utilization and flavor formation in milk, as well as genes providing acid and bile resistance. No genes coding for virulence traits or pathogenicity factors were detected. Chromosome and plasmids were packed with insertion sequence (IS) elements. Plasmids were also abundant in genes encoding heavy metal resistance traits and plasmid maintenance functions. Technologically relevant phenotypes linked to plasmids, such as the production of plantaricin C (pLL441-1), lactose utilization (pLL441-2), and bacteriophage resistance (pLL441-4), were also identified. The absence of acquired antibiotic resistance and of phenotypes and genes of concern suggests *L. plantarum* LL441 be safe. The strain might therefore have a use as a starter or starter component in dairy and other food fermentations or as a probiotic.

## 1. Introduction

*Lactiplantibacillus plantarum* is a Gram-positive, non-motile, non-spore-forming, microaerophilic and mesophilic lactic acid bacterium (LAB) found in several fermented feeds and foods of animal and plant origin, including dairy products, meat and sausages, silage, vegetables, olives, and sourdoughs [1]. *L. plantarum* grows well in raw food materials, shows proteolytic and lipolytic activities, and produces bacteriocins and other antimicrobials, properties for which strains of this species have been tested as a starter, adjunct, or protective culture [2,3]. *L. plantarum* can also be found on animal and human mucosas, such as those of the oral cavity, gastrointestinal tract, and vagina [4]. Strains showing adherence to intestinal cells and/or with certain beneficial properties, e.g., antimicrobial, antigenotoxic, or antimutagenic (among others), are marketed as probiotics [5].

*L. plantarum* can grow in diverse environments (and is thus referred to as “nomadic”), a reflection of its evolution towards generalist rather than niche specialization [6]. This is made manifest by the long genomes of its different strains, which are the longest among the typical LAB species [7], the presence of so-called “lifestyle adaptation islands” [8], and the common presence of extrachromosomal molecules (plasmids) that might aid in the dynamic and flexible adaptation of *L. plantarum* to different environmental conditions [9].

Plasmids are double-stranded, self-replicating extrachromosomal DNA molecules that occur in prokaryotic and eukaryotic microorganisms. They are mostly circular and non-essential but can help microbes grow better on particular nutrients or survive longer under unfavorable conditions [10]. They can, however, also result in a metabolic burden by diverting cellular energy and resources into plasmid-guided replication and expression [11]. It has been estimated that more than 50% of the plasmids are mobile (either conjugative or mobilizable) and, as an adaptive response to environmental changes, can be spread between species and strains occupying the same niche [12,13]. Indeed, an adaptation of LAB to the dairy environment has been thought to be a major driver of plasmid-encoded, milk-associated traits (lactose metabolism, proteolysis of caseins, bacteriophage resistance, and bacteriocin and exopolysaccharide production) [14]. This may be particularly true for species such as *Lactococcus (Lc.) lactis* and *L. plantarum*, for which large plasmid complements (the plasmidome) have repeatedly been reported [9,15,16,17].

Dairy *L. plantarum* is a member of a group of mesophilic LAB referred to as ‘non-starter lactic acid bacteria’ (NSLAB) [18]. These microorganisms are not native components of the milk microbiota and enter into dairy products as contaminants from cheesemaking tools, additives, and/or the manufacturing and ripening environments. However, *L. plantarum* and other NSLAB biotypes become dominant in most cheese types during ripening and are thought to have a role in the formation of taste and aroma compounds [19]. A complete phenotypic and genetic characterization of *L. plantarum* strains is required, however, for their reliable and safe use in food fermentations. After phenotypic testing, genome sequencing and analysis have become the gold standard for the genetic characterization of microorganisms, providing genetic support for phenotypic traits, plus insights into their full genetic potential [7,20]. As accessed in October 2022, the Genomes Online Database (GOLD) contained more than 1000 genomes for *L. plantarum* strains, although only 138 were complete.

*L. plantarum* LL441, a dairy LAB strain, was isolated as a dominant member of the microbiota of Cabrales cheese, a traditional Spanish, starter-free cheese made from raw milk [21]. It has shown strong β-galactosidase activity but weak caseinolytic capacity; consequently, it grows slowly in milk [21]. It produces a plasmid-encoded lantibiotic bacteriocin (plantaricin C) that inhibits several Gram-positive pathogenic and spoilage bacteria [22]. It is also reported to have plasmid-borne genes for a heterodimeric β-galactosidase (LacLM) [23]. The plasmid complement of the wild *L. plantarum* LL441 strain was soon presumed to be large [22,23], although the precise number of plasmids and their sizes remained uncertain. Genome sequencing using short-read (Illumina) technology recently suggested the presence of at least six plasmids, of which the largest (pLL441-1, which was completely characterized) harbors the locus for plantaricin C [24]. However, neither the actual number of plasmids nor other plasmid-associated traits had been determined. Further, completing the genome sequence would provide deeper knowledge of the LL441 biochemical/technological potentiality.

Beyond producing a lantibiotic bacteriocin that could be utilized as a natural preservative in dairy, *L. plantarum* LL441 had been shown to dominate the NSLAB cheese population during ripening [25], suggesting the strain might be of interest as a starter or adjunct culture or for use in other biotechnological applications. The present work reports the genome sequencing of *L. plantarum* LL441 using a long-read (Pac-Bio) sequencing technique and a hybrid short-read/long-read genome assembly. This allowed the bacterial chromosome and plasmidome of LL441 to be fully deciphered. Genome analysis identified genetic features that correlated with different phenotypic traits, including growth and the production of volatile compounds (VOCs) in milk. The importance of plasmid-encoded traits for *L. plantarum* LL441 with respect to growth in milk and dairy environments is also discussed.

## 2. Results and Discussion

### 2.1. Phenotypic Characterization of LL441

Table 1 summarizes the main phenotypic features recorded for *L. plantarum* LL441. Although at different rates, the strain was able to utilize 20 carbohydrates or polyalcohols on the API-50 CHL strip. The API-ZYM system revealed strong activity of LL441 for enzymes involved in the utilization of carbohydrates (β-galactosidase, α-glucosidase, β-glucosidase, and N-acetyl-β-glucosaminidase).

Strong enzyme activities involved in the release of amino acids from peptides (leucine arylamidase and valine arylamidase aminopeptidases) were also measured. Moderate activity was noted for alkaline phosphatases. In contrast, no activity was recorded for trypsin-like, α-chymotrypsin-like, and β-glucuronidase enzymes. Some bacterial serine proteases with trypsin-like and α-chymotrypsin-like activities have been associated with adverse physiological processes such as blood clotting and inflammation [26]. Bacterial β-glucuronidases have been implicated in the enterohepatic circulation of various compounds such as toxins, hormones, drugs, and carcinogens [27]. LAB β-glucuronidases have, however, also been shown to release aglycones from dietary glycosides, a property that might be beneficial [28].

The minimum inhibitory concentration (MIC) of 16 antibiotics to LL441 (Table 1) reflected the typical resistance-susceptibility pattern for *L. plantarum* [29]. The strain was resistant to high concentrations of vancomycin (MIC > 256 µL^−1^). This agrees with *L. plantarum* lacking the target for this antibiotic, the dipeptide D-Alanine(Ala)-D-Ala at the stem terminus of the pentapeptides in the cell wall peptidoglycan, which is substituted by D-Ala-D-Lac (Lactic acid) [30]. The MICs for all other antibiotics were equal to or fell below the established resistance-susceptibility cut-offs for species of the *L. plantarum*/*L. pentosus* group [31].

LL441 did not produce biogenic amines (tyramine, β-phenylethylamine, histamine, or putrescine) from the precursor compounds tested. In contrast, a small amount of gamma-aminobutyric acid (GABA) was detected in the supernatant of cultures supplemented with 5 mM monosodium glutamate (0.16 ± 0.05 mM). However, given the threshold for GABA production established by Redruello et al. [32] (>0.64 mM), the strain was considered a GABA non-producer.

LL441 grew slowly in semi-skimmed UHT milk at 32 °C, with the largest population reached about 32 h of culture (7.66 log10 cfu mL^−1^). The lowest pH was recorded at about 50 h (6.36 ± 0.01). Twenty VOCs with a relative concentration of >1% were identified by SPME-GC/MS (Table 2). Some of these were odor compounds naturally present in milk or formed during the UHT heat treatment [33]. Nevertheless, nine new compounds were detected during growth of LL441 in milk, and the concentration of five others increased significantly; the majority of compounds were organic acids produced by fermentation (acetic and butyric acids) and free fatty acids (FFAs) (caproic, caprylic, capric, and lauric). In addition, a significant increase in some ketones derived from FFAs (or the appearance of new ones) was detected. FFAs contribute greatly to flavor formation in fermented milk and cheeses, not only directly but also as precursors of methyl ketones, secondary alcohols, straight-chain aldehydes, lactones, esters, and S-thioesters [34,35]. The production of acetic acid (providing a pungent and vinegary flavor) and 2-butanone (which provides a sweet and fruity flavor) has already been attributed to *L. plantarum* when co-cultured with yogurt starters [36].

### 2.2. General Features of LL441 Genome

In previous work, genome sequencing of LL441 using Illumina short-read technology recorded 3,124,603 bp organized into 170 contigs [24]. Genome analysis identified 3017 genes, of which 2935 were predicted to be coding sequences. In the present work, LL441 was subjected to a new round of genome sequencing using long-read PacBio sequencing technology [37]. Thereupon, hybrid genome assembly of short- and long-read sequences provided the complete genome sequence of LL441, which consisted of eight circular contigs: one corresponding to the complete bacterial chromosome, plus seven different plasmids (pLL441-1 to pLL441-7).

The genome was composed of 3,195,259 bp with a GC content of 44.5%. It contained 3183 coding sequences, of which 2956 were located on the bacterial chromosome and the remaining 227 on the plasmids. The coding sequences were assigned to 232 subsystems by the RAST server. Twenty-four open reading frames (ORFs) (20 on the chromosome and 4 on plasmids) appeared disrupted by mutations and were deemed pseudogenes. Similar numbers of pseudogenes have recently been reported in other *L. plantarum* strains of dairy origin [38]. It should be noted that the genome contained >150 ORFs encoding proteins showing similarity to integrases, recombinases, and transposases of several IS elements (IS3, IS5, IS6, IS30, IS256, IS118, etc.); two-thirds of these ORFs were on the chromosome and one third on the plasmids. Some IS sequences appeared to be truncated, suggesting they might no longer code for any active protein. However, trans-complementation by functional components at other positions in the genome remains a possibility. The nucleotide identity shown by complete or partial copies of IS elements might endow the LL441 genome with a high degree of plasticity. It may have also hindered the proper assembly of the genome sequence following short-read sequencing.

Five canonical ribosomal clusters (16S, 23S, 5S) were found scattered throughout the chromosome, with tRNA-Ala-TGC and tRNA-Ile-GAT inserted within two clusters between the genes coding for the 16S and the 23S rRNA molecules. An extra copy of a 5S rRNA-encoding gene was found in the vicinity of one of these clusters. Seventy tRNA-encoding genes were also identified. Pairwise digital DNA-DNA hybridization (dDDH) between the LL441 genome and those of selected type strains from the *L. plantarum* group showed maximum identity values of 90.7% (confidence interval [CI] 88.4-92.5), 88.7 (CI 86.2–90.7), and 63.0 (CI 60.1-65.8), respectively, against the genomes of *L. plantarum* subsp. *plantarum* ATCC 14917^T^, *L. plantarum* subsp. *plantarum* DSM 13273^T^ (formerly known as *Lactobacillus arizonensis*), and *L. plantarum* subsp. *argentarotensis* DSM 16365^T^ (formerly known as *Lactobacillus argentarotensis*). This confirmed the taxonomic assignment of LL441 as *L. plantarum* subsp. *plantarum*. The same assignment was inferred from ortho average nucleotide identity (orthoANI) values and phylogenomic analysis between LL441 and type strains of all other species of the *L. plantarum* group (Appendix A).

Phage-related ORFs scattered on the chromosome, plus an entire prophage sequence, have been previously reported in the LL441 genome [24]. In the present work, analysis of the complete genome sequence confirmed the presence of this prophage and identified a second. The two prophages embraced DNA segments of 48.8 and 59.8 kbp and harbored contiguous ORFs showing extensive homology, respectively, to the *Lactobacillus* phages phiAT3 (NC_005893.1) and Sha1 (NC_019489.1), both of the *Siphoviridae* family. This page content agrees well with the results of a recent pan-genome analysis of different *L. plantarum* strains [9].

### 2.3. Plasmidome of LL441

Although some 13.3% of *L. plantarum* genomes contain no plasmids at all, others contain up to 14 [9]. For *L. plantarum* LL441, six to eight plasmid bands had already been detected in cell extracts [21,22,25]. Six ORFs coding for replication proteins belonging to different plasmid families were also known after short-read sequencing of the genome [24]. The complete sequence of pLL441-1 -the plasmid harboring the plantaricin C locus- had also been determined in the latter work. In the present study, the hybrid assemblage revealed eight circular contigs, i.e., the bacterial chromosome plus seven plasmid molecules (pLL441-1 to pLL441-7) (Figure 1). The sizes of these contigs were, respectively, 55,314 bp (the same as that determined by Flórez and Mayo [24]), 48,660, 40,749, 31,725, 9181, 8844, and 8686 bp. The circularity of the bacterial chromosome and four of the plasmids was demonstrated by PCR and sequencing of the amplicons. Surprisingly, only three of these contigs (pLL441-5, pLL441-6, and pLL441-7) were detected as plasmids by the PlasmidFinder program. Genes coding for replication proteins were clearly identified in all (pLL441-1 contained two) but one of the plasmids (pLL441-6). The absence of a *dso*-like sequence suggests the latter plasmid does not belong to the rolling circle (RC) family. Further, the presence of a large AT-rich region, similar to that reported for other plasmids from Gram-positive bacteria [39], suggests pLL441-6 follows a theta-type replication mode. Despite the absence of a recognizable replication protein, pLL441-6 harbored an ORF coding for a DUF536 domain-containing protein, which might account for this function. An identical gene has recently been reported in a plasmid isolated from cheese [40]. Figure 2 shows the phylogenetic relationships of LL441 plasmid-associated proteins to those of other *L. plantarum* plasmids and to those of well-known plasmids representative of the main plasmid replication families. Although both RC- and theta-type replication mechanisms have been found in *L. plantarum* plasmids, most replicating proteins of *L. plantarum* plasmids, including most of LL441, show no homology to replication proteins of representative plasmid families of these two types (Figure 2). Therefore, the plasmids of *L. plantarum* deserve further attention given our lack of knowledge regarding their genetic organization, replication and control, and the limitations of present gene expression platforms and tools for this species [41].

No CRISPR-Cas systems were detected in the LL441 genome, although iteron repeats in the coding region of the pLL441-1.1 replication protein were recognized as repeats and spacers by the CRISPRCasTyper program. As some authors suggest, the absence of CRIPR-Cas systems agrees well with the large plasmid complement of LL441 and the presence of two integrated phages [42]. Nonetheless, a gene encoding a bacteriophage abortive infection protein of the AbiH family was identified in pLL441-3. Abi systems are characterized by a normal start of infection followed by an interruption of the phage development leading to the release of few or no progeny particles and to the death of the infected cell [43]. A homologous protein to that in LL441 provides total resistance to small isometric-headed bacteriophages and a reduction in the plating efficiency of prolate-headed bacteriophages to *Lc. lactis* [44]. Finally, one-third of the largest plasmid, pLL441-1, possessed OFRs coding for conjugation functions, although the transfer of this (or indeed of any other LL441 plasmid) by conjugation was not observed.

### 2.4. Lactose Metabolism in LL441

Lactose is metabolized by LAB via the Leloir or tagatose-6-phosphate pathways. Usually, however, only genes for the Leloir pathway are found in *L. plantarum* [45]. Indeed, a gene coding for a heterodimeric β-galactosidase (LacLM) of plasmid origin has been cloned previously from LL441 [23]. However, loss of the β-galactosidase-coding plasmid only reduced the activity of this enzyme; congruently, the *lacLM* genes had been found duplicated in LL441 and other *L. plantarum* strains [46]. With this background, it was not surprising that the present genome analysis of LL441 detected a β-galactosidase-encoding gene of the *lacZ*-type, plus two of the *lacLM*-type.

The gene coding for the LacZ β-galactosidase (*lacZ*) was detected in the bacterial chromosome in the neighborhood of the *lacLM* genes coding for a heterodimeric LacLM enzyme (Figure 3A). All these genes were found in a long (82.2 kbp) DNA segment flanked by inverted copies of an IS6501 element (Figure 3A). The structure of this long segment resembled that of a “lifestyle adaptation island” [8]. Within this segment, genes involved in the utilization of fructose, cellobiose, N-acetylglucosamine, N-acetylmuramic acid, and other glucosides were also identified. At least one other island of about 61 kbp harboring genes encoding carbohydrate transporters and enzymes involved in the utilization of maltose, maltodextrin, and sucrose, was also noted. A second copy of the *lacLM* genes was also encountered in pLL441-2 (Figure 3B). Indeed, a 7.5 kbp DNA fragment harboring the β-galactosidase genes was found to be duplicated (identical nucleotide sequence) in the chromosome and in pLL441-2. In addition to the heterodimeric enzyme, this segment also included genes coding for an α-galactosidase and the lactose permease LacS (PTS) (Figure 3B). Both the chromosomal and plasmidic β-galactosidase gene copies were surrounded by genes encoding lactose/galactose permeases and other genes involved in the transport or utilization of carbohydrates other than lactose (Figure 3). It is tempting to speculate that the evolutionary pressure of the milk environment has forced duplication and shuffling of genes between the chromosome and plasmids, giving rise to an increase in the activity of key enzymes acting on lactose by enhancing their copy number and/or relaxing control.

### 2.5. Genetic Potential for Flavour Production

No gene encoding a caseinolytic proteinase required for rapid growth in milk, such as that found in other LAB species [47], was identified in the LL441 genome, which agrees well with the strain growing slowly in this medium. However, a complex repertoire of genes involved in protein (proteases) and peptide (endo-, amino- and carboxypeptidases) degradation and subsequent amino acid catabolism and flavor formation (aminotransferases, transaminases, and dehydrogenases) was identified (Appendix A). Nonetheless, genes encoding pivotal LAB components to grow and produce aroma compounds in milk, such as *pepA* (glutamyl aminopeptidase), *pcp* (pyrrolidone-carboxyl peptidase), and *araT* (aromatic amino acid aminotransferase gamma) were not detected [47,48].

Except for a metallo-endopeptidase of the ImmA/IrrE family and a class III alcohol dehydrogenase located in pLL441-1, all other genes were found on the chromosome, which strongly suggests this gene collection belongs to the core genome.

Diacetyl (2,3-butanedione) and acetoin (3-hydroxy-2-butanone) -creamy and buttery flavour compounds formed from pyruvate- are pivotal sensory compounds in many dairy products [49]. Two metabolic pathways for the production of diacetyl/acetoin have been proposed in LAB. In the first, seen in *Lc. lactis* subsp. *lactis* biovar *diacetylactis* and *Leuconostoc* spp., lactose, and citrate are co-metabolized to produce an excess of pyruvate, which is converted into diacetyl. In the second, the direct synthesis of diacetyl from acetyl-CoA is postulated. However, the diacetyl synthase enzyme has never been isolated from (or its gene identified in) any LAB species [50]. In the genome of LL441, no genes involved in citrate metabolism were identified beyond those coding for the citrate lyase complex (Appendix A). These enzymatic composite splits citrate into acetate and oxaloacetate, compounds that enter the tricarboxylic acid cycle, the main energy pathway of cells.

Lipolysis relates to the hydrolysis of triglycerides and other lipids by the action of esterases and lipases, leading to the liberation of free fatty acids (FFA) [51]. FFA and its derived metabolites contribute significantly to the development of specific flavors in dairy products [52]. A few genes encoding esterases/lipases and phosphoesterases, together with a large collection of genes (>18) putatively encoding non-specific hydrolases, were identified in the LL441 genome (Appendix A). How these enzymes are related to the VOCs profile determined by SPME/GC remains to be determined.

### 2.6. Genome Analysis and Other Functionalities

Some LAB species and strains are thought to provide beneficial health effects to consumers by the production of bioactive metabolites either in food during fermentation or in the gastrointestinal tract after consumption [53]. The presence in the LL441 genome of the GAD gene (*gadB*), and a gene encoding a putative proton/glutamate symporter, has already been reported [54]. A GAD-like gene is present in the genome of most *L. plantarum* strains. *gadB*, however, was not associated with the gene encoding the typical glutamate/γ-aminobutyrate antiporter (*gadC*) found in strong GABA-producing LAB [55]. In addition, the two genes identified in LL441 (*gadB* and the gene encoding the symporter) were not contiguous on the chromosome, suggesting they are not coordinately regulated. This, plus the fact that LL441 and all other 17 *L. plantarum* strains assayed by Valenzuela et al. [54] were deemed to be GABA non-producers, suggests the biological function of the proton/glutamate symporter might not be involved in H^+^/GABA exchange.

LL441 has been recorded as producing tiny amounts of conjugated linoleic acid (CLA) from linoleic acid (LA) (Valenzuela et al., unpublished). Conversion of LA into CLA requires the concerted action of hydrolase and isomerase enzymes [56]. Indeed, many hydrolase-encoding genes were detected in the genome of LL441, as well as three contiguous ORFs, the deduced proteins of which showed complete amino acid identity with an isomerase complex (Cla-dh, Cla-dc, and Cla-er) known to be involved in the synthesis of CLA by *L. plantarum* AKU 1009a [57].

Some years ago, a list of genes involved in probiotic activities and adaptation factors contributing to the best-documented health-promoting actions of lactobacilli in the gastrointestinal tract was proposed [58]. This list has been recently updated [9]. The probiotic activities include direct inhibitory effects, as well as the synthesis of antipathogenic, epithelium barrier-preserving, and immunomodulatory molecules, while the adaptation factors comprise determinants for stress resistance, metabolism in the host, and adherence to the gut mucosa [58]. With respect to the direct inhibitory effects, the plantaricin C locus was reported in LL414 in a previous study [24]. Although the inhibitory spectrum of plantaricin C was deemed limited to a few Gram-positive species [22], it might modulate the development of some microbial populations under the stressful conditions of the gut. In the present work, no further bacteriocin operons were noted. With respect to the adaptation factors, a panoply of encoding genes was identified (Appendix A). However, the absence of a genetic makeup devoted to the synthesis of key colonization and persistence features, such as pili and fimbriae, exopolysaccharide production, biofilm formation, and a lack of pivotal genes such as *bshA* (bile salt hydrolase) and *xylA* (xylose isomerase) [58], was also noticed. These genes and their associated features are strain-specific and have only been identified in a minority *of L. plantarum* genomes [9]. Altogether, these findings suggest that LL441 may be well able to survive in harsh environmental conditions, such as that of the gastrointestinal tract, while it possesses a moderate capacity to colonize it.

### 2.7. Genome Analysis and Safety

Although the presence of 29 genes coding for efflux pumps were noted in the LL441 genome, searches in CARD and ResFinder databases identified no acquired antibiotic-resistance genes. Neither did the latter database identify any chromosomal point mutation resulting in antibiotic resistance. Analysis of the D-Ala-D-Ala ligase gene showed a conserved phenylalanine (F) residue at position 260 of its deduced amino acid sequence (…KLGAIDVPKT DTFYDYNNKF VDASGVTFEL PVELPADLTK…). This residue forms part of the ligase active site and correlates with vancomycin resistance in lactobacilli [29]. Comparison with the genes of *Enterococcus faecalis*, the closest species in VFDB, revealed only nine genes somewhat similar in the genome of LL441. Some of these were housekeeping genes involved in cellular functions (e.g., protein turnover), while others coded for adherence proteins or capsular components. Moreover, at the protein level, they showed low homology (35–60%) to the *E. faecalis* counterparts. No genes coding for cytolysins, gelatinase, or hyaluronidase enzymes were detected. Similar results were obtained after searching in the Victor’s database. Thus, the LL441 genome would appear to contain no gene giving rise to safety concerns.

In agreement with a negative phenotype for the production of biogenic amines, no genes coding for decarboxylases that might act on the precursor amino acids assayed were detected. However, two genes encoding carboxylase-like proteins (that of GAD and a lysine decarboxylase-family protein) were identified. The gene for the latter was associated with another coding for an L-O-lysylphosphatidylglycerol synthase, an enzyme involved in the synthesis of the major bacterial membrane phospholipid [59,60], suggesting the carboxylase may participate in the synthesis of the cell wall.

## 3. Materials and Methods

### 3.1. Culture Conditions

*L. plantarum* LL441 was originally isolated from a 60-day-old Cabrales cheese in 1985 by Mayo et al. [21]. Unless otherwise stated, *L. plantarum* LL441 was grown in static MRS broth (Oxoid, Basingstoke, UK) or on 2% MRS agar plates in aerobiosis at 32ºC for 24-48 h. Growth in UHT semi-skimmed milk (CAPSA, Siero, Spain) was assayed at 32 °C as described by Rodríguez et al. [61].

### 3.2. Phenotypic Characterization

*L. plantarum* LL441 was subjected to a battery of biochemical tests. These included the examination of its carbohydrate fermentation profile using the API50 CHL system (bioMérieux, Montalieu-Vercieu, France) and the determination of its enzyme activities using the semi-quantitative API-ZYM system (bioMérieux). Further, as previously reported [61], its antimicrobial resistance-susceptibility profile to 16 antibiotics using broth microdilution and EULACB1 and EULACB2 plates (ThermoFisher, Waltham, MASS, USA) was also determined. Biogenic amine production from precursor amino acids and derived compounds (tyrosine, histidine, agmatine, and ornithine) was assessed as reported elsewhere [62]. Gamma-aminobutyric acid production (GABA) from monosodium glutamate (5 mM) was quantified, as reported by Valenzuela et al. [54].

### 3.3. Production of Volatile Compounds (VOCs) in Milk

The production of VOCs in milk was assayed in triplicate using a headspace (HS) solid-phase micro-extraction (SPME)/gas chromatographic (GC) technique after growth in UHT milk (CAPSA, Siero, Spain) at 32 °C for 48 h. The resulting samples were stored at −20 °C until analysis. Frozen control (unfermented) and fermented samples were later thawed, and 5 mL was placed in 20 mL screw cap SPME vials (Agilent, Santa Clara, CA, USA). As an internal standard (IS), 40 µL of styrene (0.25 ppm) was added. All vials were sealed using a PTFE/silicone liner septum (Agilent) and equilibrated at 60 °C for 10 min with pulsed agitation at 500 rpm for 5 s using a PAL RSI 120 device (CTC Analytics, Zwingen, Switzerland). VOC was extracted using ARR11-DVB-120/20 DVB/PDMS fibers (CTC Analytics) which were exposed to the headspace above the samples at 60 °C for 20 min. After removal, they were placed in the inlet of an Agilent 8890 gas chromatograph equipped with an HP-INNOWax (30 m × 250 um × 0.25um) column (Agilent), using a split/splitless injector in splitless mode, and desorbed for 1 min at 250 °C. The temperature was set at 35 °C and held for 0.5 min, then increased at a rate of 6.5 °C/min up to 165 °C and thereafter at 20 °C/min up to 260 °C. Helium was used as the carrier gas at a constant flow rate of 1.4 mL min^−1^. An Agilent MSD 5977B mass spectrometer running in single quadrupole mode was used as a detector. The ion source temperature was 230 °C, and the interface temperature was 280 °C. The MS mode was electronic ionization (70eV) with the mass range scanned between 35 and 250 amu.

Mass spectra were normalized by the IS area and compounds identified by matching against the NIST 2014 mass spectral library of the National Institute of Standards and Technology (https://chemdata.nist.gov/) (Accessed on November 2022), following the Institute’s guidelines. The threshold for compound integration was 1% of the relative response.

### 3.4. Genome Sequencing

The short-read sequencing of *L. plantarum* LL441 was reported in a previous paper [24]. In the present work, total genomic DNA was extracted from an overnight culture and purified using the GenElute Bacterial Genomic DNA kit (Sigma-Aldrich, St. Louis, CA, USA). Long-read sequencing was then performed using a PacBio Sequel II SMRT sequencer at FISABIO (Valencia, Spain). After sequencing, a hybrid assembly approach (combining short and long reads) was applied. For this, short reads were downloaded from GenBank (BioProject accession number PRJNA318575), and after trimming long reads with TrimGalore (https://www.bioinformatics.babraham.ac.uk/projects/trim_galore/) (Accessed on 21 July 2022), short and long read sequences were assembled with Unicycler v0.5.0 [63]. Assembly errors were polished using Pilon [64] and Racon [65].

### 3.5. Genome Analysis

Whole genome sequence data were used to assess the phylogenetic relationship between the sequenced strain and type strains of the *L. plantarum* group via digital DNA-DNA hybridization (dDDH) [66] and orthologous average nucleotide identity (orthoANI) analysis [67]. The genome of *L. plantarum* LL441 was analyzed using the most updated version of publicly-available online tools and software programs. Annotation and analysis were initially conducted using PATRIC services (https://www.patricbrc.org/) (Accessed on 21 July 2022). Plasmid-encoded ORFs and their deduced proteins were manually compared against the GenBank database using the Blast tools (https://blast.ncbi.nlm.nih.gov/Blast.cgi) (Accessed on 21 July 2022). Antibiotic resistance and virulence genes were investigated by genome comparison against sequences in Resfinder (https://cge.cbs.dtu.dk/services/ResFinder/) (Accessed on July 2022) [68] and CARD (https://card.mcmaster.ca/) (Accessed on 21 July 2022) [69], and VFDB (Virulence Factor Database; http://www.mgc.ac.cn/VFs/) (Accessed on 21 July 2022) [70] and Victors (http://www.phidias.us/victors/) (Accessed on 21 July 2022) [71] databases, respectively. The genome sequence was searched for plasmids using PlasmidFinder (https://anaconda.org/bioconda/plasmidfinder) (Accessed on 21 July 2022) [72]. Ribosomally-synthesized and post-translationally modified peptides and bacteriocins were sought using BAGEL 4 (http://bagel4.molgenrug.nl/) (Accessed on 21 July 2022) [73]. CRISPR-Cas systems were searched for using the CRISPRCasTyper v.1.6.4 online tool (https://crisprcastyper.crispr.dk) (Accessed on 21 July 2022) [74]. Prophage Hunter software program was used to search the genome sequence for prophages (https://pro-hunter.genomics.cn/) (Accessed on 21 July 2022) [75].

## 4. Conclusions

In this work, *L. plantarum* LL441 was subjected to a range of phenotypic tests, including carbohydrate utilization, enzyme activity, resistance to antibiotics, and growth and VOC production in milk. By combining short- and long-read sequencing results, the whole genome of *L. plantarum* LL441 was assembled and was seen to consist of eight circular contigs corresponding to the bacterial chromosome plus seven plasmids. Three of these plasmids harbor key functions for adaptation to growth (lactose-utilizing genes) and the ability to compete (bacteriocin production, bacteriophage resistance) in the milk environment. The absence of any phenotype or gene of safety concern suggests *L. plantarum* LL441 might be used with confidence as a starter or adjunct culture in dairy and food fermentations or as a probiotic.

## Figures and Tables

**Figure 1 ijms-24-00605-f001:**
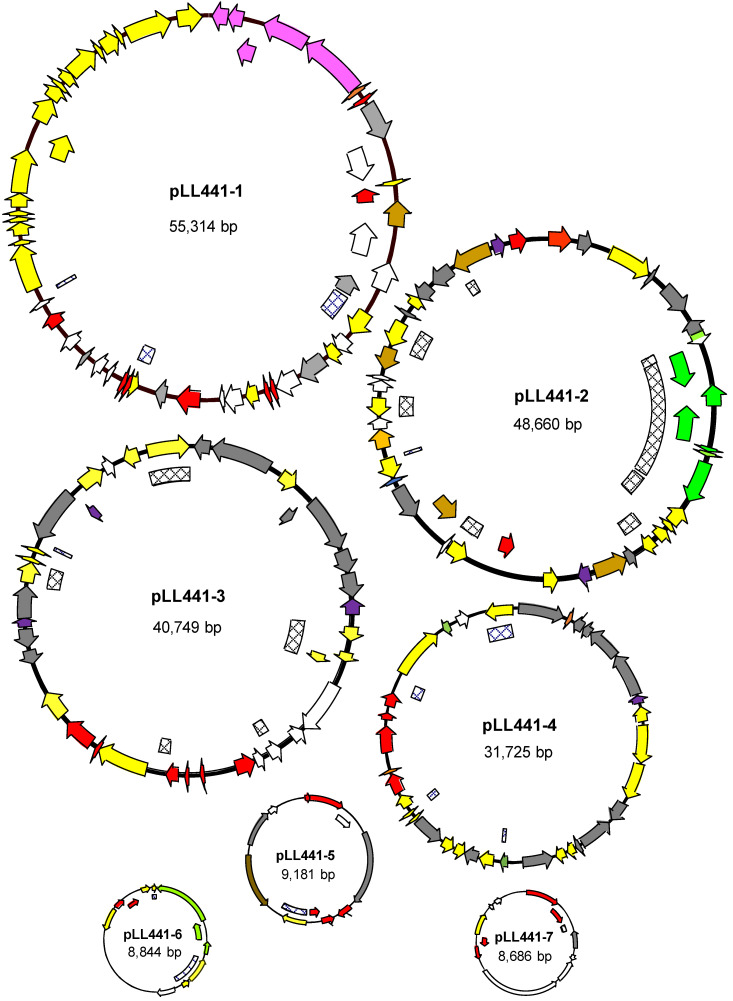
Genetic maps of the seven plasmids of *L. plantarum* LL441 (pLL441-1 to pLL441-7). Arrows indicate the position, direction, and length of the ORFs. Color key: red, genes encoding plasmid replicating and maintenance functions; yellow, IS elements and genes involved in mobilization; green, genes involved in the carbohydrate metabolism; purple, plantaricin C-associated genes; brown, genes encoding other or unknown functions; white, hypothetical ORFs. Stripped boxes indicate the sequences are repeated on the genome. Sizes of the plasmids are relative but not drawn to scale.

**Figure 2 ijms-24-00605-f002:**
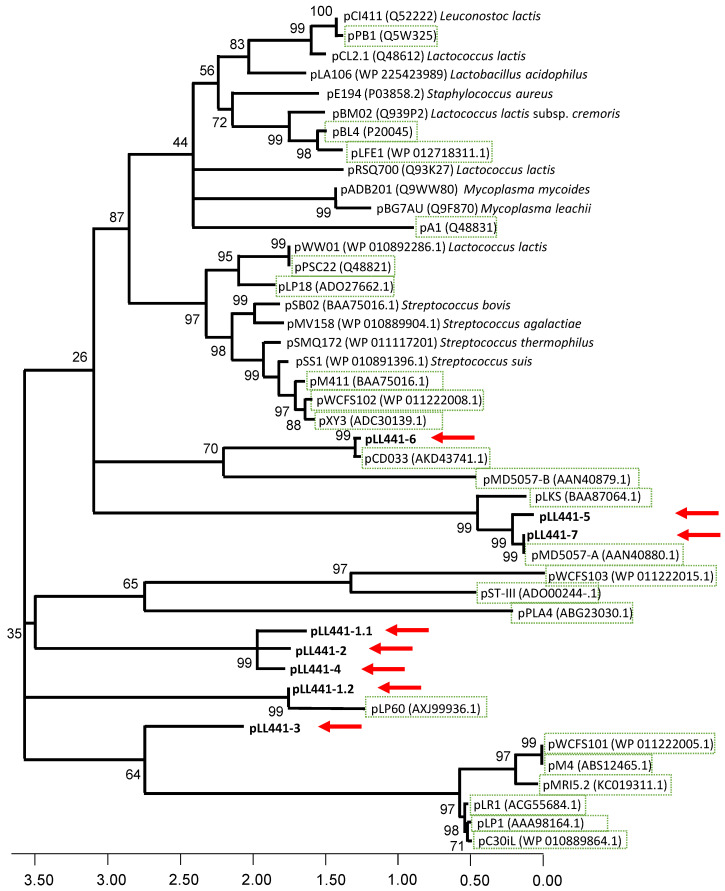
Dendrogram obtained with the distance matrix using the neighbor-joining method of the deduced amino acid sequences of RepB proteins from the seven plasmids of LL441 (pLL441-1 to pLL441-7) and those of other *L. plantarum* plasmids (boxed in green) and representative plasmid of different rolling-circle and theta-type plasmid families from other bacteria. Evolutionary distances were measured as the mean number of amino acid differences per site. The red arrows point toward the position of the *L. plantarum* LL441 plasmids.

**Figure 3 ijms-24-00605-f003:**
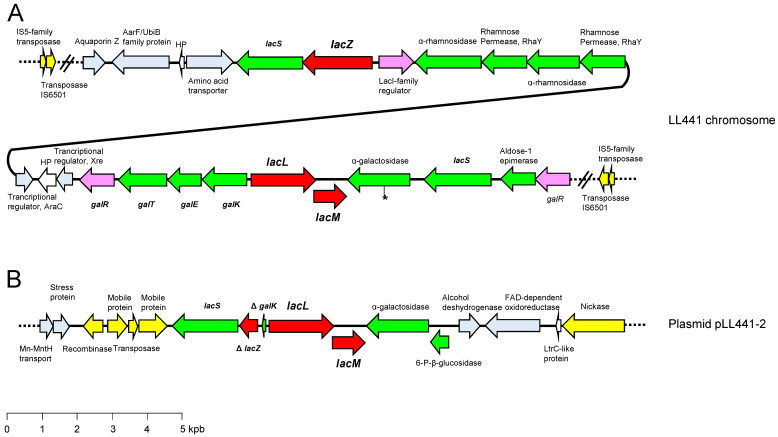
Genetic organization of the lactose gene clusters found in the chromosome (**A**) and plasmid pLL441-2 (**B**) of the *L. plantarum* LL441 genome. Colour code: in red, genes coding for β-galactosidases, *lacZ,* and *lacLM*; in green, genes for carbohydrate transport and metabolism systems; in light purple, genes encoding regulatory proteins; in yellow, open reading frames (ORFs) encoding transposases or mobilization proteins; in light blue, other genes; in white, genes encoding hypothetical proteins (HP). The asterisk in the chromosomal copy of the α-galactosidase ORF denotes a point mutation disrupting the gene. The *rhaY* and *araC* genes, linked by a black line, are contiguous in the chromosome. The dotted lines at the end of the clusters and the crossed lines indicate the continuity of the regions.

**Table 1 ijms-24-00605-t001:** Phenotypic characterization of *L. plantarum* LL441.

Utilization of Carbohydrates		Enzyme Activity		Antibiotic Resistance	
Carbohydrate	Degree of Utilization ^a^	Enzyme	Activity ^b^	Antibiotic	MIC ^c^
D-ribose	++++	Alkaline phosphatase	0	Gentamicin	1
D-galactose	++++	Esterase (C 4)	<1	Kanamycin	16
D-glucose	++++	Esterase lipase (C 8)	<1	Streptomycin	16
D-fructose	++++	Lipase (C 14)	0	Neomycin	1
D-mannose	++++	Leucine arylamidase	>40	Tetracycline	32
D-mannitol	++++	Valine arylamidase	30	Erythromycin	0.25
D-sorbitol	++++	Cystine arylamidase	<1	Clindamycin	1
N-acetylglucosamine	+++	Trypsin	0	Chloramphenicol	8
Amygdaline	++	α-chymotrypsin	0	Ampicillin	1
Arbutine	+++	Acid phosphatase	5	Penicillin G	4
Esculine	++++	Naphthol-AS-BI-phosphohydrolase	10	Vancomycin	>256
Salicine	+++	α-galactosidase	0	Quinupristin-dalfopristin	1
D-cellobiose	++++	β-galactosidase	>40	Linezolid	4
D-maltose	++++	β-glucuronidase	0	Trimethoprim	0.25
Lactose	++++	α-glucosidase	20	Ciprofloxacin	16
Mellibiose	++++	β-glucosidase	30	Rifampicin	2
D-sucrose	++++	N-acetyl-β-glucosaminidase	30		
D-threhalose	++	α-mannosidase	0		
Gentiobiose	++	α-fucosidase	0		
D-turanose	++				

^a^ Only carbohydrates utilized are reported; ^b^ Enzyme activity (nM of substrate hydrolysed); ^c^ MIC, minimum inhibitory concentration (in µg mL^−1^).

**Table 2 ijms-24-00605-t002:** Volatile compounds (VOCs) produced by *L. plantarum* LL441 in milk.

Compound	Control Unfermented Milk	Milk Fermented by LL441
**Acids**		
Acetic acid	-	2.64 ± 0.39
Butyric acid (butanoic acid)	-	5.54 ± 1.51
Caproic acid (hexanoic acid)	6.58	50.45 ± 16.62
Caprylic acid (octanoic acid)	13.61	83.93 ± 24.54
Perlargonic acid (nonanoic acid)	-	1.66 ± 0.36
Capric acid (n-decanoic acid)	22.06	65.87 ± 14.71
Caproleic acid (9-decenoic acid)	-	5.06 ± 1.57
Lauric acid (dodecanoic acid)	4.11	9.56 ± 2.17
**Ketones**		
2-Heptanone	10.55	9.21 ± 5.32
2-Dodecanone^a^	-	3.73 ± 2.15
2-Undecanone	4.14	5.47 ± 1.35
2-Tridecanone	-	2.61 ± 0.70
2-Nonanone	8.74	9.43 ± 1.95
4-Methyl-2-hexanone ^a^	3.23	14.30 ± 8.25
6-pentyl-2H-pyran-2-one	2.17	3.44 ± 0.53
**Lactones**		
γ-Dodecalactone	1.78	1.69 ± 0.23
δ-Dodecalactone	1.95	2.52 ± 0.35
**Other**		
1,3-bis(1,1-Dimethylethyl)benzene (sulfur compound)	-	2.29 ± 1.33
4-Piperidinepropanoic acid, 1-benzoyl-3-(2-chloroethyl) ^a^ (alcohol)	-	1.43 ± 0.42
Tridecyl alcohol tri(oxyethylene) ethanol ^a^ (alcohol)	-	2.66 ± 0.89

-, not detected; ^a^ These compounds were tentatively identified with a match factor < 600.

## Data Availability

The PacBio genome sequencing data of *L. plantarum* LL441 was deposited in the NCBI database under BioProject number PRJNA913822 (CP114874- CP114881).

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
