# Peer review of "Phenotypic and Safety Assessment of the Cheese Strain Lactiplantibacillus plantarum LL441, and Sequence Analysis of its Complete Genome and Plasmidome"

_ijms, 2022, doi:10.3390/ijms24010605_

Round 1

Reviewer 1 Report

Reviewer comments and suggestions

The authors in this study demonstrated the phenotypic typing and complete genome analysis of cheese strain Lactiplantibacillus plantarum LL441. The authors reported that the genome of LL441 included eight circular molecules, the bacterial chromosome and seven plasmids (pLL441-1 through pLL441-7), ranging in size from 8.7 to 53.3 kbp. 

Genome analysis revealed massive arrays of genes involved in carbohydrate utilization and flavour formation in milk, as well as genes providing acid and bile resistance. Authors also reported the production of plantaricin C (pLL441-1), lactose utilization (pLL441-2), and bacteriophage resistance (pLL441-4) which were relevant phenotypes linked to the plasmid complement. Finally, they concluded that the strain could be used a as a starter or starter component in dairy and other food fermentations, or as a probiotic.

Overall, the manuscript was well written that could be accepted in IJMS. Few concerns/comments needed to be explained/modified. 

  1. Line 12-13 What does it mean, please check the sentence
  2. Line 57-58 Explore these studies
  3. Line 66-67 Why it is so
  4. Line 81-84 it was already reported with a reference and presented in the main abstract so how the authors could explain the novelty of this study
  5. Table 2 how this compound checked that mentioned in the legend part, even though methods they included elsewhere.
  6. Line 339 Please check a typo error
  7. Line 380-384 please avoid long sentences, and modify
  8. Comments for section 3.5 it would be nice if the authors could write a few points on the software they used
  9. Journal name should be italics, and the year should be in bold numerical, please try to correct all.

Author Response

Reviewer 1

The authors in this study demonstrated the phenotypic typing and complete genome analysis of cheese strain Lactiplantibacillus plantarum LL441. The authors reported that the genome of LL441 included eight circular molecules, the bacterial chromosome and seven plasmids (pLL441-1 through pLL441-7), ranging in size from 8.7 to 53.3 kbp.

Genome analysis revealed massive arrays of genes involved in carbohydrate utilization and flavour formation in milk, as well as genes providing acid and bile resistance. Authors also reported the production of plantaricin C (pLL441-1), lactose utilization (pLL441-2), and bacteriophage resistance (pLL441-4) which were relevant phenotypes linked to the plasmid complement. Finally, they concluded that the strain could be used as a starter or starter component in dairy and other food fermentations, or as a probiotic.

Overall, the manuscript was well written that could be accepted in IJMS. Few concerns/comments needed to be explained/modified.

Line 12-13 What does it mean, please check the sentence

For the sake of clarity, the sentence has now been rewritten. It currently reads:

LL441 utilized a large range of carbohydrates and showed strong activity of some carbohydrate-degrading enzymes.

Line 57-58 Explore these studies

The two articles have been carefully read and, as a consequence, the sentence has been slightly modified to be more precise, and the second referred article has been replaced by the original article, as follows:

It has been estimated that more than 50% of the plasmids are mobile (either conjugative or mobilizable) and, as an adaptive response to environmental changes, can be spread between species and strains occupying the same niche [12,13].

  1. Smillie, C.; Garcillán-Barcia, M.P.; Francia, M.V.; Rocha, E.P.C.; de la Cruz, F. Mobility of plasmids. Microbiol Mol Biol Rev. 2010, 74, 434-452. doi: 10.1128/MMBR.00020-10.

Line 66-67 Why it is so

We apologize; the meaning of the sentence was obscure and did not reflect the idea we wanted to transmit. The sentence has been modified, and it reads now as:

These microorganisms are not native components of the milk microbiota, and enter into dairy products as contaminants from cheesemaking tools, additives, or the manufacturing and ripening environments.

Line 81-84 it was already reported with a reference and presented in the main abstract so how the authors could explain the novelty of this study

Absolutely; thank you. We should have stated at this point what is still lacking, which would have explained the need for further sequencing. A couple of short sentences have been added to state these facts, as follows:

However, neither the actual number of plasmids nor other plasmid-associated traits had been determined. Further, completing the genome sequence would provide deeper knowledge of the LL441 biochemical/technological potentiality.

Table 2 how this compound checked that mentioned in the legend part, even though methods they included elsewhere.

We agree with the reviewer in that stating incubation conditions in the tables we incurred an unnecessary repetition, which should have been avoided. Temperature, time of incubation, and the methodology used have now been removed from the title and footnotes of all tables.

Line 339 Please check a typo error

The typo has been corrected; thank you.

Line 380-384 please avoid long sentences, and modify

In the revised version, this long sentence has now been split in three, as follows:

L. plantarum LL441 was subjected to a battery of biochemical tests. These included the examination of its carbohydrate fermentation profile using the API50 CHL system (bioMérieux, Montalieu-Vercieu, France) and the determination of its enzyme activities using the semi-quantitative API-ZYM system (bioMérieux). Further, as previously reported [61], its antimicrobial resistance-susceptibility profile to 16 antibiotics using broth microdilution and EULACB1 and EULACB2 plates (ThermoFisher, Waltham, MASS, USA) was also determined.

Comments for section 3.5 it would be nice if the authors could write a few points on the software they used

We appreciate the suggestion raised by the reviewer. However, the bunch of software programs and online tools used are all found in the public domain, and most of them are well-known within the scientific community involved in genome sequencing and analysis of prokaryotes. This makes it very easy to reach and consult; all of them do have a short introductory front page with their essentials. Therefore, we believe that describing functionalities, strengths and/or weaknesses, here, wouldn´t add much to the manuscript while enlarging the text of the heading (which is already long).

Journal name should be italics, and the year should be in bold numerical, please try to correct all.

The list of references has now been updated following the Instructions for the Authors of the IJMS journal.

Reviewer 2 Report

The manuscript describes a detailed phenotypic and genomic characteristics of L. plantarum LL441. The manuscript is well written and the methods used are appropriate. I have just two minor suggestions:

- Please add the acc. no. of the genome sequence.

- Why exactly this strain is so important. Why exactly this strains was selected for sequencing? Please, add a short explanation at the end of the introduction.

Author Response

Reviewer 2

The manuscript describes detailed phenotypic and genomic characteristics of L. plantarum LL441. The manuscript is well written and the methods used are appropriate. I have just two minor suggestions:

- Please add the acc. no. of the genome sequence.

The PacBio sequence data of L. plantarum LL441 have been submitted to the NCBI (GenBank) database under the BioProject accession number PRJNA913822; this code has now been included in the manuscript.

- Why exactly this strain is so important? Why was exactly this strain selected for sequencing? Please, add a short explanation at the end of the introduction.

The reasons for selecting this specific strain might have been scattered throughout the manuscript. We agree with the reviewer that stating this clearly from the very beginning is paramount in order to make the reader understand the possible impact of the work. A couple of sentences have been added to the last paragraph of the introduction to stress these facts, as follows:

Beyond producing a bacteriocin that could be employed as a natural preservative in dairy, L. plantarum LL441 had been shown to dominate the NSLAB population of cheese during ripening [25], suggesting the strain might be of interest as a starter or adjunct culture, or for use in other biotechnological applications.